# Green Consumer Behavior of Sports Enthusiasts on TikTok—An Analysis of the Moderating Effect of Green Concern

**DOI:** 10.3390/bs14040285

**Published:** 2024-03-30

**Authors:** Yuan-Fu Lee, Chen-Yueh Chen, Ya-Lun Chou, Yi-Hsiu Lin

**Affiliations:** 1Doctoral Program for Transnational Sport Management and Innovation, College of Management, National Taiwan Sport University (NTSU), Taoyuan 333325, Taiwan; 1091703@ntsu.edu.tw (Y.-F.L.); chenchenyueh@ntsu.edu.tw (C.-Y.C.); 2Master Program of Sport Facility Management and Health Promotion, National Taiwan University, Taipei 10617, Taiwan

**Keywords:** green brand image, green word of mouth, customer stickiness, purchase intention, green concern, TikTok

## Abstract

The short-form video platform TikTok has become highly popular. This study explores how professional sports teams can effectively leverage short-form videos to promote green values such as environmental conservation and sustainable development, thereby capturing user attention and enhancing user engagement. This study primarily aimed to investigate the effects of a green brand image on green word of mouth (WOM), customer stickiness, and consumer purchase intention, with further examination regarding the moderating effect of green concerns on these relationships. Few studies have explored the presence of professional sports teams on TikTok, particularly in the context of green issues. Accordingly, this study adopts a novel method to develop specific and actionable recommendations for professional sports teams who have a presence on social media. Additionally, via the application of the Stimulus–Organism–Response theory, this study explains how the green brand image presented by professional sports teams on TikTok influences the interactive relationships among green WOM, customer stickiness, and consumer purchase intention. This study recruited 600 individuals who were either fans of the Taipei Fubon Braves, which is a team in Taiwan’s professional basketball league P.LEAGUE+, or fans of other teams. After a confirmatory factor analysis, structural equation modeling was used to test the hypotheses. The results indicate positive correlations in all tested paths. The green concern of the Taipei Fubon Braves’ fans moderated the relationship between green WOM and purchase intention; however, this moderating effect was not identified among the non-Taipei Fubon Braves fans. These findings introduce innovative concepts to the field of marketing, contributing substantially to both practical applications and academic research.

## 1. Introduction

Consumers are increasingly concerned about the environment [1], and businesses have responded to these concerns [2,3]. Numerous studies have demonstrated the correlation between the willingness to purchase and the green behaviors of customers [4,5,6]. Thus, many businesses aim to establish a green brand image to increase the distinctiveness of their environmentally friendly products [7,8,9]. Rahmi et al. [10] demonstrated that consumers are more willing to purchase from brands with a green image.

The role of social media in promoting green concerns and establishing brand identity has been widely acknowledged. Studies have demonstrated the profound influence of social media strategies on brand image [11,12,13,14]. In the sports industry, the brand image of corporate teams is closely linked to brand impressions, consumer engagement, and their emotional attachment to specific teams. This connection highlights the critical role of brand image in the success of professional sports [15]. Ristevska-Jovanovska [16] emphasized the criticality of social media as a tool for promotion, communication, and interaction among fans, athletes, and sports organizations. Utilizing social media in sports marketing can effectively generate brand awareness and foster supporter engagement [17]. Therefore, establishing a green brand image becomes crucial for the management of professional teams, and is the central focus of this study.

A green brand image refers to consumer perceptions and impressions of a brand’s commitment to sustainability; it encompasses the awareness, views, and potential behavioral responses of consumers toward the brand [18]. Bashir et al. [19] highlighted that establishing a green brand image requires businesses to actively communicate their environmental commitments, promote eco-friendly products and services, and implement environmental actions. These actions include reducing one’s ecological and carbon footprint and engaging in sustainability efforts [20]. According to Majeed et al. [21], when consumers have a positive image of a brand, they are more inclined to exhibit positive behavioral intentions. In the context of sports, a sports team’s brand image is closely associated with brand impressions, consumer engagement, and the emotional connection of consumers to a specific team. Thus, the brand image of professional sports teams is crucial for their success [15].

TikTok, an emerging and popular social media platform, has achieved notable success through its innovative features and unique business model [22]. Its key differentiator from other platforms is its advanced “algorithm”. Central to TikTok’s appeal is this algorithm, providing users with tailored information and experiences [23]. Lu and Lu [24] indicated that users typically do not actively search for specific content on TikTok; instead, they rely on the platform’s recommendation algorithm to discover videos, which contrasts the active content search often required for YouTube. Users are drawn to TikTok’s easy-to-consume, knowledge-sharing video format.

In response to TikTok’s success, several companies have launched their own short-form video applications, such as Instagram Reels and YouTube Shorts, inspired by TikTok’s interface and functionality [25]. TikTok’s greatest advantage is its amalgamation of powerful features from various social media platforms, resulting in a unique combination of strengths. Primarily, it stands out as a visual media platform, excelling in user engagement. Its algorithmic recommendation system is unparalleled, offering personalized experiences to users. Additionally, its mobile interface is tailored for smartphone cameras, and simplifies content creation and sharing [26].

Managing social media is pivotal for brand image [27]. With technological advancements, the significance of social media has grown due to its wide reach, and professional sports have recognized its value in connecting with fans [28]. TikTok has demonstrated efficacy in social media marketing, particularly for sports organizations [29]. During the COVID-19 hiatus, professional athletes turned to TikTok, frequently posting short videos to engage with fans and maintain player–fan interactions during competition pauses [30]. Sports organizations have also increased their presence on TikTok [31]. In 2019, the European football club Liverpool FC became the first Premier League team to join TikTok, using the platform to engage new online audiences, particularly younger fans [32]. Although TikTok’s adoption in the sports industry may not have been widespread initially, its utility as a social media platform was established prior to the COVID-19 pandemic [33].

Businesses not only promote their goods or services [34,35,36,37], but also emphasize their commitment to sustainability in their TikTok marketing. In particular, the inherent design of social media platforms to encourage virality helps companies disseminate their content [38]. Through social media marketing, businesses not only profit, but also communicate their commitment to corporate social responsibility.

Word of mouth (WOM) marketing has been transformed, becoming increasingly central as a result of social media marketing [38,39]. Sadat [40] demonstrated that brand image mediates the relationship between WOM and purchase intention. Agarwal and Teas [41] also indicated that brand image influences purchase intention, customer satisfaction, and customer loyalty by influencing perceived value. Maintaining a positive brand image also contributes to the enhancement of customer enthusiasm and stickiness [42]. Moreover, with the increased use of social media by consumers, their sensitivity to and identification with the green brand image, conveyed by professional sports teams via TikTok short videos, have also increased. This phenomenon strengthens the brand stickiness of these sports teams [43].

The aim of this study was to primarily investigate the effects of having a green brand image on green word-of-mouth (WOM), customer stickiness, and consumer purchase intention, with further examination focusing on the moderating effect of displaying green concern on these relationships. Moreover, this study explores how promoting a green brand image on TikTok can help professional sports teams promote green WOM, enhance customer stickiness, and reinforce purchase intention. Additionally, the present study examines the moderating effect of green concern on the relationships among variables in order to comprehensively understand the interactive relationship between a green brand image and WOM. The present study fills a research gap regarding professional sports teams on TikTok, with particular regard to environmental sustainability in such a context.

Specifically, this study investigates the effects of a professional sports team’s green brand image, as conveyed through engagement on TikTok, on green WOM, customer stickiness, green concern, and consumer purchase intention. The present study proposes eight research hypotheses that are then tested using structural equation modeling (SEM); in these hypotheses, green concern is posited as a moderating variable.

## 2. Literature Review and Hypothesis Development

### 2.1. Stimulus–Organism–Response Theory

The present study applies the Stimulus–Organism–Response (S–O–R) theory to investigate how a professional sports team’s green brand image on TikTok influences the interactive relationships among green WOM, customer loyalty, and consumer purchase intention. The S–O–R theory has been widely used to study the psychology and behavior of Internet users, including the effects of various external environmental factors [44]. Introduced by Mehrabian and Russell [45], the S–O–R theory pertains to the relationship between environmental stimuli (Stimulus), psychology (Organism), and behavior (Response).

In the present study, the “Stimulus” pertains to the green brand image promoted by professional sports teams on TikTok, including their video content, brand information, and sustainability practices. Hu and Jasper [46] revealed that the brand image of a store significantly influences consumer willingness to shop, thereby demonstrating the utility of the S–O–R theory for understanding brand image. “Organism” refers to the cognitive and emotional responses of social media users, which are key factors given that marketing is an inherently psychological endeavor [47,48]. In the present study, green WOM and customer stickiness are regarded as the organisms, where cognitive and emotional responses can influence user perceptions of a green brand and their willingness to support the brand [49,50]. “Response” is represented by consumer purchase intention, which indicates whether consumers are willing to support or purchase products related to a green brand. Studies have successfully applied the S–O–R theory to explain how external stimuli can affect the internal psychological states of consumers, thus influencing their behavior on online shopping platforms [51]. Therefore, the present study aims to validate the relationships among variables through an S–O–R model.

In addition, the S–O–R theory offers valuable insights into sports marketing and brand promotion. For instance, Li et al. [52] conducted empirical research on the Chinese Basketball Association (CBA) sport sponsors, TCL Group and Li-Ning Company, investigating the influence of sports sponsorship motives on consumer purchase intentions. They examined consumer attitudes as a mediating factor, establishing a theoretical framework based on the S–O–R theory, combining empirical research with theoretical construction. This framework clarified how sports sponsorship motives influence consumer purchase intentions from the perspective of the S–O–R theory, providing essential insights into the application of the S–O–R theory and motivational theory in sponsorship marketing. It also lays the groundwork for validating the relationships between variables in the current study via the use of the S–O–R theory model.

### 2.2. Green Brand Image

Park et al. [53] defined brand image as the interconnections between information nodes in the consumer’s memory and brand nodes. They differentiated brand image into the functional, symbolic, and experiential dimensions. Building on this definition, Chen [54] introduced the concept of a “green brand image”, defining it as “consumers’ perceptions of the brand involving environmental commitments and green concern”. 

Alamsyah et al. [55] emphasized that a business’s green brand image contributes to the establishment of trust among consumers regarding the business’s environmental commitments. This trust can influence the willingness to purchase and purchase decisions of consumers, thereby increasing the market share of a business. When conducting marketing activities on TikTok, businesses can strengthen their green brand image by publishing creative content related to environmental conservation.

According to Sipari [56], through the environmental content of TikTok, in its capacity as a video-sharing platform, the awareness of sustainability among its users can be increased. Green brands can leverage TikTok’s features to disseminate information regarding environmental knowledge, lifestyles, and products, thereby establishing a professional image that reflects environmental protection. Trendafilova et al. [57] also emphasized the role of social media in promoting and shaping the environmental sustainability efforts of professional sports teams and influencing the individual and group attitudes, beliefs, and behaviors towards these organizations.

Furthermore, environmental issues have become a global concern and focus of academic research [58]. Therefore, actively promoting environmental activities and fulfilling corporate responsibility can not only enhance a brand’s green reputation, but can also help to create a positive environmental image in the minds of the consumers.

### 2.3. Green Brand Image and Green WOM

A green brand image refers to the image that a business establishes in the minds of consumers regarding its environmental commitments and concerns [59]. Consumer perceptions of a brand’s environmental and sustainable efforts can be defined as a green brand image, which closely relates to brand attitude [60]. A highly effective channel for influencing consumer attitudes is WOM communication [61]. In the minds of consumers, WOM plays a crucial role in shaping brand image, helping a business to generate consumer interest in its products and prompting purchases. Research has revealed that WOM significantly affects brand image [62].

When professional sports teams post content on TikTok that emphasizes their commitment to the environment, they are likely to elicit positive responses from consumers regarding their environmental efforts, leading to favorable WOM. Thus, the present study proposes Hypothesis H1-1 as follows:

**Hypothesis (H1-1).** The green brand image of professional sports teams on social media positively predicts green WOM.

### 2.4. Green Brand Image and Purchase Intention

According to Rahmi et al. [10], a positive green brand image has a positive effect on green purchase intention. This finding suggests that businesses with a positive green brand image are more likely to influence customers to purchase their environmentally friendly products or services than businesses without such an image. Conversely, a negative green brand image may reduce the willingness of consumers to purchase [63]. When consumers perceive a business’s green brand image favorably, they are more inclined to buy its products [64]. A green brand image also influences the willingness of consumers to make repeat purchases, their positive recommendations to others, and their willingness to pay more [65]. When professional sports teams present information regarding environmental conservation and sustainable development on TikTok in order to strengthen their green brand image, consumers become more inclined to support these teams, as reflected in their increased purchase intention. Therefore, the present study considered these factors to assess how the green brand image of professional sports teams on TikTok influences the purchase intention of consumers. Thus, the present study proposes Hypothesis H1-2 as follows:

**Hypothesis (H1-2).** The green brand image of professional sports teams on social media positively predicts purchase intention.

### 2.5. Green Brand Image and Customer Adherence

Beddoe-Stephens [66] introduced the concept of “stickiness”, defining it as the ability to maintain user engagement and attract them to revisit a given website [67]. Chen [54] contended that businesses should focus on establishing a positive green brand image to enhance the stickiness and market competitiveness of green brands [68]. Therefore, how brands communicate their green brand image is crucial for creating positive interactions between these brands and consumers, and for building green brand value through the joint efforts of both consumers and brands.

TikTok videos are highly engaging and are thus suited to marketing [69,70,71], particularly as it relates to a green brand image. This enables a business to quickly attract the attention of users and strengthen its brand image in the minds of consumers, thereby influencing customer stickiness in relation to a brand. In the literature, few studies have investigated how a green brand image affects customer stickiness. Thus, the present study proposes Hypothesis H1-3 as follows:

**Hypothesis (H1-3).** The green brand image of professional sports teams on social media positively predicts customer stickiness.

### 2.6. Green WOM and Consumer Purchase Intention

WOM can be defined as the sharing of information regarding a product among potential and past customers [72,73,74]. WOM is a crucial source of information that influences consumer behavior [75,76]. Furthermore, having positive green WOM facilitates the building of consumer trust, thereby enhancing the willingness of consumers to purchase green products [77]. Research has indicated a positive association between positive WOM intention and purchase relationships [77,78].

Sheikhalizadeh [79] examined the influence of sports advertising on consumer purchase intentions via WOM on social media platforms, focusing on how consumers mediate this relationship. The research emphasized the substantial influence of WOM on the purchase intentions of consumers. In the context of purchasing sports products, consumers are often influenced by WOM from others, thereby affecting their purchase intentions.

When professional sports teams effectively communicate their green brand image on TikTok, they strengthen the positive effect of green WOM on consumer purchasing behavior, thereby increasing the likelihood of consumers choosing to support these teams’ green products or services. Thus, the present study proposes Hypothesis H2 as follows:

**Hypothesis (H2).** The green WOM of professional sports teams on social media positively predicts purchase intention.

### 2.7. Customer Stickiness and Consumer Purchase Intention

Lin et al. [80] indicated that stickiness has a crucial effect on customer purchase intention. Zhang et al. [81] discovered a positive correlation between customer engagement and stickiness on company social media platforms. When consumers are highly engaged in social media, they tend to frequently visit and linger on social media platforms [82]. If customer engagement has a positive effect on stickiness, then it also has a positive influence on consumer repurchase intention [83]. Additionally, scholars have reported a positive correlation between the social media usage intensity and purchase intention of individuals (or fans) [84]. Therefore, we infer that, on TikTok, the customer stickiness achieved by professional sports teams plays a crucial role in consumer purchase intention. When fans feel a strong emotional connection to the team they support, they tend to be more inclined to purchase products or services related to the team [85]. Thus, the present study proposes Hypothesis H3 as follows:

**Hypothesis (H3).** The customer stickiness of professional sports teams on social media positively predicts consumer purchase intention.

### 2.8. Green Brand Image, Green WOM, and Consumer Purchase Intention

When consumers share positive reviews about a business’s environmental protection measures on the Internet, they generate green WOM. These WOM interactions can strengthen the green brand image of a business, creating a positive feedback loop. Studies have confirmed a positive relationship between a green brand image and green WOM [62,86,87,88,89]. Furthermore, brand image can create reputation and brand prestige, thereby influencing consumer willingness to purchase the products of a brand [90]. Research has demonstrated that a green brand image significantly influences the purchase intention of consumers [91,92,93]. Finally, green WOM involves consumers sharing their positive opinions regarding the environmental aspects of products, which can influence their friends, relatives, and colleagues, thereby motivating these people to purchase green products [94].

In summary, green WOM may moderate the relationship of the green brand image of professional sports teams on TikTok with consumers’ purchase intention, reinforcing the effects of brand image on purchase intention. Thus, the present study proposes Hypothesis H4-1 as follows:

**Hypothesis (H4-1).** Green WOM moderates the relationship of the green brand image of professional sports teams on social media with consumer purchase intention.

### 2.9. Green Brand Image, Customer Stickiness, and Purchase Intention

Businesses strengthen their brand image and consumer purchase intention by engaging in positive interactions and communications with consumers [95]. The positive effect of a green brand image on consumer purchase intention has been verified [92,93]. Given that a stable brand image has a positive effect on customer enthusiasm and adherence [42], the stability of the brand image helps enhance customer enthusiasm and stickiness.

The findings of a study suggest that achieving brand exposure on social media platforms can help a business to enhance its brand image [96]. Social media platforms, such as TikTok, have become an essential aspect of brand promotion, and professional sports teams are increasingly posting short-form video content on TikTok in order to increase their brand exposure and attract a young audience [97]. Montag et al. [98] argued that professional sports teams must create attractive content to engage young users, form emotional connections with a young audience, and enhance the image and awareness that consumers have of their brand. Research has indicated that frequent interactions between a brand’s fan page and consumers encourage consumers to maintain a stable relationship with the brand, thereby increasing the likelihood of repeat purchases, WOM propagation, and increased brand influence [99]. That is, these close, frequent interactions positively affect a brand through increasing consumer exposure and strengthening customer relationships [100]. According to Lin et al. [80], the length of time users spend on a website is closely related to their purchase intention, with adherence having a significant effect on this relationship. Thus, the present study proposes Hypothesis H4-2 as follows:

**Hypothesis (H4-2).** Customer stickiness moderates the relationship of the green brand image of professional sports teams on social media with consumer purchase intention.

### 2.10. Green Brand Image, Green WOM, and Green Concern

A green brand image can be defined as a set of perceptions that consumers have regarding a brand with respect to the brand’s environmental commitments and concerns [54]. Furthermore, a green brand image can also be understood as the feelings and relationships that consumers form in relation to their commitment to and concern for the environment [101]. Zhang et al. [102] defined green WOM as the positive evaluations made by consumers regarding the environmental nature of a product or business. Their study emphasized the connection between green WOM and brand image, suggesting that consumers with strong green concerns are more likely to consider whether a business takes actual environmental actions, rather than only considering its brand image. Research has consistently demonstrated a positive relationship between brand image and WOM [39,40,89,103].

The present study proposes that green concerns are a moderating variable in the relationship between the green brand image and green WOM of professional sports teams on TikTok. Understanding the reactions of highly environmentally conscious individuals to the green brand image and WOM of professional sports teams on TikTok can provide deeper insights into the effects of environmental awareness on a green brand image and WOM in specific contexts. Thus, the present study presents Hypothesis H5 as follows:

**Hypothesis (H5).** Green concern moderates the relationship between a green brand image and green WOM on professional sports teams’ social media.

## 3. Methodology and Measures

### 3.1. Sampling and Collection of Data

This study received ethical approval from the Institutional Review Board of National Taiwan University. The Taipei Fubon Braves basketball team, a part of the Taiwan P.LEAGUE+ professional basketball league, was chosen as the subject for this study. This selection was based on Fubon Financial’s strong emphasis regarding employee well-being, its considerable involvement in sports, and its efforts in promoting sustainable development principles. Fubon Financial has been a long-standing sponsor of the Taipei Marathon, Taiwan’s largest annual running event. Since 2018, it has reached over 100,000 audience members and participants, sponsoring more than 23 sports events, including the LPGA Taiwan Championship, tennis tournaments, marathons, weightlifting competitions, and youth baseball events [104]. Notably, Fubon Financial is the only corporation in Taiwan that owns both a professional baseball team (Fubon Guardians) and a semi-professional basketball team (Taipei Fubon Braves).

Fubon Financial, as the parent company of the Taipei Fubon Braves, entered into a naming partnership with the Taipei City Government, starting from the 2019–2020 season. The team officially participated in competitions under the name Taipei Fubon Braves. In July 2020, the Taipei Fubon Braves were instrumental in the establishment of the P.LEAGUE+, marking a crucial milestone in the development of professional basketball and league sports in Taiwan. Recently, Fubon Financial launched the Fubon Earth Team green advertising campaign, reflecting its commitment to sustainable development.

Green concerns and sustainable development are increasingly prominent. However, to the best of our knowledge, other Taiwanese leagues such as T1 and SBL have not prominently engaged in environmental advocacy or sustainable development-related activities or promotions on TikTok. Globally, research focusing on the role of TikTok in addressing sports sustainability concerns is limited. Therefore, this study delved into the motivations and perceptions of fans, analyzing their attitudes and behaviors toward sustainable development. The results of this study are expected to enhance our understanding of individual’s concerns regarding sustainable development challenges.

The study participants comprised two distinct groups. The first group consisted of fans of the Taipei Fubon Braves, who were recruited from an online community on the LINE social messaging platform after their status as fans of the team was confirmed via a questionnaire survey. The second group consisted of fans of P.LEAGUE+ teams other than the Taipei Fubon Braves (i.e., non-Taipei Fubon Braves fans). These fans were also recruited through online communities on LINE after their status as non-supporters of the Taipei Fubon Braves and fans of other teams was confirmed via the questionnaire survey. This study investigated the attitudes and behaviors of Taipei Fubon Braves basketball team fans after they viewed the Fubon Earth Team advertisement on TikTok. Additionally, the study aimed to identify differences in perceptions and behaviors between fans of the Taipei Fubon Braves (supporters of the target team) and fans of other teams (nonsupporters). Additionally, this study explored the extent to which fans are willing to actively participate in and support green concerns, regardless of their team allegiance.

This study employed a questionnaire survey method, hosted on the SurveyCake platform. This survey was disseminated via link to research participants who met the eligibility criteria. As an incentive, participants who completed the questionnaire were rewarded with LINE POINTS, equivalent to cash. The participant recruitment process, conducted through convenient sampling, resulted in a total of 600 completed questionnaires. Given the widespread use of LINE in Taiwan, with approximately 90.7% of the population being users [105], this social media platform was selected for distributing the questionnaire to ensure effective sampling. Respondents completed the questionnaire through LINE, with links to relevant TikTok videos included within the survey. Participation being limited to members of online communities on LINE could result in sampling bias, because it excludes individuals not using this platform. This limitation might restrict the comprehensiveness of the sample and challenge the generalization of the study findings to the broader population. Future studies may consider recruitment via various social media platforms to ensure a more diverse sample and to mitigate potential biases.

The initial step of the research process involved identifying suitable participants. Subsequently, participants were required to watch a short video produced by Fubon Financial Holding, titled ‘Fubon Earth Team’. This video contains elements related to the green brand image, such as environmental protection and sustainability values. By viewing this video, participants could form a brand image aligning with these values. The content is designed to stimulate discussion and sharing among participants, thus generating green WOM and attracting attention to the brand. Additionally, the green brand image portrayed in the video is anticipated to enhance consumer loyalty and influence purchase intentions. The video is accessible through the TikTok application for playback. After confirming that participants have watched the video, they were directed to complete the questionnaire. The questionnaire comprised sections gathering the basic information of participants and assessing their perceptions of the green brand image, green WOM, customer loyalty, intentions to purchase, and green concerns. Completing the questionnaire marked the end of the survey process.

### 3.2. Measurements

In the present study, the measured demographic variables were the perception of a green brand image, green WOM, customer stickiness, green concern, and consumer purchase intention. We modified the existing definitions and measurement methods for these variables in the literature to fit the context of our study. Specifically, the green brand image, green WOM, customer stickiness, green concern, and consumer purchase intention were measured using modified versions of the scales developed by Chen [54], Zhang et al. [102], Yang and Lee [106], Zhang et al. [81], and Hien and Nhu [107], respectively. Notably, the modifications made to the green brand image scale were based on the recommendations of Cretu and Brodie [108] and Padgett and Allen [109], and those made to the customer stickiness scale were based on the recommendations of Zhang et al. [81]. All scale items were rated on a 7-point Likert scale, with endpoints ranging from 1 (strongly disagree) to 7 (strongly agree).

## 4. Results

PROCESS 4.2 software (Model 7) was used for statistical analysis, and R software (2022.07.2) was used for SEM analysis.

### 4.1. Demographic and Measurement Model Results

The descriptive statistics revealed a gender distribution for the sample that closely aligns with the broader TikTok user base in Taiwan. Among participants, men accounted for 46% of the sample, which is slightly lower than the 51% representation of men among all TikTok users in Taiwan. Women comprised 54% of the sample, which is marginally higher than their 49% representation in the overall Taiwanese TikTok user demographic. This lack of extreme distribution in terms of gender is noteworthy. In the age category, the sample had a notable representation of young adults, aligning well with the demographic trends of TikTok users in Taiwan. Individuals aged between 20 and 30 years accounted for 24% of the sample, matching their 24% representation among the total TikTok user base in Taiwan. Those aged between 31 and 40 years formed 45% of the sample, closely paralleling the 44% representation of this age group among TikTok users in Taiwan. These figures highlight the predominance of the young adult demographic among TikTok short video viewers, reflecting current digital marketing trends that focus on younger audiences. In the income category, those earning between TWD 20,001 and TWD 40,000 constituted 43% of the sample, identical in proportion to TikTok’s Taiwan userbase. Individuals earning between 40,001 and 60,000 represented 34% of the sample compared with 32% among TikTok users in Taiwan. This similarity suggests that the sample captures a significant segment of TikTok users with considerable purchasing power, which is particularly relevant for environmentally friendly products. Regarding the educational level, 78% of the participants had a college degree compared with 75% of TikTok users in Taiwan. This correlation may indicate a higher awareness and knowledge regarding environmentally friendly products among the sample, potentially influencing their purchase intentions. Regarding marital status, married individuals made up 52% of the sample compared with 48% among TikTok users in Taiwan, whereas unmarried individuals accounted for 48%, which is slightly less than the 52% making up TikTok’s Taiwan demographic. This slight variance indicates possible differences in attitudes toward environmentally friendly branded products or services based on marital status. For further information, please refer to Table 1.

The model was reliable. The Cronbach’s α for all structures ranged from 0.81 (0.86) to 0.94 (0.93), and a confirmatory factor analysis revealed that all related indicators were within an acceptable range [110] (see Table 2). The model was also valid. In particular, the model displayed discriminant validity. Specifically, per Fornell and Larcker’s [111] recommendation, the average variance extracted (AVE) for each construct was greater than its squared correlation with other constructs, and all AVE values exceeded 0.50 (see Table 3). Finally, no common method bias was detected in a Harman’s single-factor test [112] and pre-rotation principal component analysis. Specifically, the results indicated that the variance of the Taipei Fubon Braves fans was 46.33%, and that of the non-Taipei Fubon Braves fans was 48.82%, which were both lower than the requisite maximum of 50% for total variance [113].

### 4.2. SEM Analysis Results

SEM analysis was then conducted (Table 4), and the results indicated a good fit. Subsequently, path analysis was conducted, and the results indicated that all paths were significant. Specifically, a green brand image displayed significant and positive relationships with green WOM, purchase intention, and stickiness; furthermore, green WOM and stickiness had significant and positive relationships with purchase intention. The results supported all hypotheses in both samples (Figure 1). The coefficient of determination (R^2^) for each latent variable in the research model was as follows: green WOM (0.58), purchase intention (0.59), and customer loyalty (0.13).

### 4.3. Moderated Mediation Analysis

Model 7 of PROCESS 4.2, provided by Hayes [114], was used to determine whether green concern moderated the effect of a green brand image on green WOM and purchase intention (H4) in both samples. Bootstrapping was performed with 5000 samples in order to generate a 95% confidence interval (CI; [115]).

Among the Taipei Fubon Braves fans, green concern had an interaction effect with a green brand image (β = 0.10, standard error [SE] = 0.04, 95% CI = [0.18, 0.02], *p* < 0.05; Table 5). Thus, green concern moderated the relationship between a green brand image and green WOM (Figure 2). The sample was segmented into high versus low levels of green concern (+1 and −1 standard deviations from the mean, respectively). A green brand image had a significant effect on green WOM in the groups, with both demonstrating high (β = 0.74, SE = 0.07, 95% CI = [0.87, 0.61], *p* < 0.001) and low (β = 0.56, SE = 0.07, 95% CI = [0.69, 0.43], *p* < 0.001) levels of green concern (Table 6). Among the non-Taipei Fubon Braves fans, no interactive effect was identified between green concern and the green brand image (β = 0.05, SE = 0.05, 95% CI = [0.16, −0.05], *p* > 0.05). These results indicate that green concern did not moderate the relationship between the green brand image and green WOM among the non-Taipei Fubon Braves fans. The differential outcomes between the two groups can be interpreted using social identity theory. This theory posits that individuals tend to affiliate with groups that reflect their values and identities, and they maintain their social identity by supporting the behaviors of these groups. Therefore, among Taipei Fubon Braves fans, supporters might be more inclined to support environmentally friendly brands that align with their values and identity. This shared identity can strengthen the influence of the green brand image on green WOM. However, nonfans might lack this sense of identification with the Taipei Fubon Braves team; therefore, they might be less responsive to the relationship between green concerns and the green brand image. These findings, particularly among the Taipei Fubon Braves fans, support H5.

## 5. Discussion

The present study explored five concepts, namely the green brand image, green WOM, customer stickiness, consumer purchase intention, and green concern, and evaluated their relationships via the use of a conceptual framework. The hypothesized positive relationships (H1–H4) are supported in both samples.

In this study, H1-1, suggesting a positive relationship between a green brand image and green WOM, was supported. Fans of the Taipei Fubon Braves exhibited a highly positive response to the team’s green brand image. This reaction may be a result of their strong emotional connection with the team. Heere and James [116] observed that fans of professional sports teams develop emotional connections, perceiving their favorite team’s symbolism as an expression of their own self-concept and identity. Therefore, by fostering these emotional connections with sports consumers, sports team managers can enhance the team’s brand assets [117]. For example, Nike’s emotional engagement with consumers is evident in products such as Flyease athletic shoes, which stemmed from a narrative involving a child with disabilities, and resulted in a design that requires no shoelace tying [118]. Khuong and Cable Car’s [119] research underscores the influence of emotional marketing on purchase decisions. Therefore, emotional advertising imagery and targeting specific emotional images are crucial for achieving higher levels of consumer purchase decisions. Moreover, loyal consumers often identify strongly with a brand image due to emotional factors [120]. This loyalty is pronounced among sports fans, who tend to establish emotional connections with their teams. When the Taipei Fubon Braves presented a positive green brand image on TikTok, it resonated more deeply with fans, thus enhancing their positive evaluations and the team’s green reputation. Conversely, non-Taipei Fubon Braves fans although not loyal, still demonstrated a favorable response to the green brand image, possibly indicating the general attractiveness of green elements to consumers. Hwang et al. [121] highlighted that consumers’ concern for the environment affects their evaluation of a brand’s sustainability commitments. Consumers are more likely to have a favorable perception of a brand when its image aligns with their self-concept [122]. Rahman et al. [62] further confirmed that a green brand image positively influences green reputations, aligning with the findings of H1-1 in this study, as well as demonstrating a positive correlation between the green brand image of a professional sports team and a green reputation on the TikTok platform.

H1-2 is supported by the results of the present study. The Taipei Fubon Braves fans, being loyal supporters of the team, might have responded more positively to the green brand image promoted by the team on TikTok. However, positive corporate initiatives can also create a strong brand image and brand assets, resulting in increased consumer purchase intention [123]. According to Grewal et al. [124], consumer impressions of product quality are closely related to brand image. When consumers are choosing between similar products, brand image is often a crucial factor, influencing their purchase decisions due to time and knowledge constraints [91]; this finding aligns with those of other studies. Among the non-Taipei Fubon Braves fans, who were not loyal to the Taipei Fubon Braves, the identified positive correlation indicates their positive response to the green brand image. Larasati and Octavia’s [125] findings suggest that a positive green corporate image has a positive effect on the loyalty of green customers. When consumers perceive a business as having a favorable green brand, they are more inclined to purchase its products [64]. The present study revealed that presenting a green brand image on TikTok not only enhanced team loyalty, but also increased the appeal of the team, highlighting the emphasis of fans on environmental consciousness. Fianto et al. [95] also emphasized the effect of brand image on the purchase intention of consumers. This result aligns with the initial hypothesis, thereby supporting H1-2.

H1-3 is supported by the results of the present study. The Taipei Fubon Braves fans might already exhibit a high level of stickiness to the team, and the positive green brand image of the team on TikTok could have further strengthened their loyalty. According to Chen [54], businesses should focus on building a strong green brand image, provide environmentally friendly products or services that meet consumer needs, and gain the trust of consumers in their environmental commitment, thereby enhancing the stickiness and market competitiveness of the green brands being promoted [68]. Stickiness is crucial for a brand because it promotes consumer loyalty and repeat purchases, and increases market share and brand value [126]. The high stickiness of the fans in the present study may be attributed to their frequent participation in team-related interactions on TikTok. The green brand image of the Taipei Fubon Braves could have resonated emotionally with these fans, increasing their alignment with the values of the team and, consequently, their stickiness. In terms of the non-Taipei Fubon Braves fans, who were not supporters of the Taipei Fubon Braves, they may not have developed long-term stickiness to the team. However, Hu et al. [127] discovered that a key factor influencing customer stickiness is the psychological responses of customers, including their perception, emotions, and behavioral responses to a platform. Scholars have indicated that corporate social responsibility (CSR) has a significant effect on the intention of consumers to use eco-friendly products from a brand [128]. A possible reason is that this group might be influenced by the Taipei Fubon Braves’ public commitment to CSR. Thus, the non-Taipei Fubon Braves fans might also have identified with this brand image. That is, the reinforcement of brand image could have played a crucial role in helping consumers to establish an emotional connection with the team. Thus, a team can design brand marketing experiences to increase consumers fondness for its brand and enhance its brand image, thereby increasing the likelihood of consumers making continual purchases [129] and establishing an emotional connection with the team. This result aligns with the initial hypothesis, thereby supporting H1-3.

The support for H2 in this study indicates that an increase in green reputation corresponds to a more positive consumer purchase intention. Relevant research has established a correlation between reputation and consumer purchase intention [77], which corroborates the findings supporting H2. Fans of the Taipei Fubon Braves, owing to their deeper emotional connection with the team, might be more influenced by the team’s reputation within their fan community or the wider public. Chaniotakis and Lymperopoulos [130] observed that consumers typically share environmental messages about products through green WOM, affecting purchasing behaviors within their social and professional networks. Similarly, Mikalef et al. [131] demonstrated that sharing product information and reviews on social media significantly influences the purchase intentions and behaviors of other potential consumers. Rimadias et al. [132] observed that active brand promotion on TikTok often stems from previous purchase experiences or recommendations to others. TikTok, as a platform, effectively disseminates product messages, thereby playing a key role in the spread of online reputation [133]. Establishing positive brand awareness on TikTok, through sharing engaging content such as product showcases and brand stories, can positively and significantly influence purchase decisions, thus enhancing green reputation [134]. In the context of relationship marketing, a positive reputation results from consumer-business relationships and consumer loyalty [135]. Therefore, a positive green reputation may reinforce the positive perceptions of the team held by fans of the Taipei Fubon Braves, thereby increasing their willingness to purchase team merchandise or attend games. For nonfans of the Taipei Fubon Braves, green reputation can be a crucial factor in their awareness and evaluation of the team. When uncertainty surrounds green products, consumers typically gravitate toward trusting and purchasing products that have received positive green reputation feedback [136]. A positive green reputation may change their impression of the team, thereby increasing their inclination to support the team through merchandise purchases or game attendance. The findings are consistent with Keller and Fay’s [137] assertion that a positive reputation engenders high levels of trust. When consumers hear others sharing positive messages about products, their propensity to make purchase decisions is heightened.

H3 is supported by the results of the present study. Lee et al. [85] demonstrated that an increase in customer engagement increases the likelihood of consumers maintaining brand stickiness, thereby affecting their willingness to repurchase. This finding aligns with the results supporting H3 in the present study. The Taipei Fubon Braves fans’ active engagement on TikTok can be assessed using the indicators proposed by Gillespie et al. [138], which include dwell time, frequency of use, and the depth of user engagement with a social platform; such an assessment can be used to evaluate their stickiness to TikTok. However, high stickiness contributes to improved interactions between customers and sellers [139]. When consumers spend more time on a platform, they are more likely to make a purchase [140]. Therefore, when Taipei Fubon Braves fans focused more on platform information, their purchasing intention was more likely to increase. The non-Taipei Fubon Braves fans might still be interested in the brand and might have developed stickiness due to the brand’s emphasis on social responsibility and sustainable development. According to Wang and Li [141], when consumers perceive that the information quality of green advertising is high, the act of emphasizing environmentally friendly attributes can enhance the persuasive effect of green advertising, thereby influencing the decisions of such consumers. Thus, our results confirm the positive effect of customer stickiness on the purchase intention of consumers.

The results pertaining to H4-1 and H4-2 confirm that green WOM and customer stickiness play moderating roles, influencing consumer purchase intention via the green brand image of professional sports teams on TikTok. Studies have reported a positive relationship between a green brand image and green WOM [62,86,87,88,89]. Given the argument that WOM communication, especially when emphasized, has a considerable effect on product evaluation [142], we believe that the WOM communicated online through dynamic and interactive media platforms, such as the Internet, can significantly influence product evaluation. This phenomenon may involve brand image and purchase intention [89], and it aligns with the results pertaining to H4-1 in the present study. These findings serve as robust evidence which supports the viewpoint that a green brand image has a positive effect on green WOM, emphasizing the relevance of a green brand image for eco-friendly products and services. That is, a green brand image can influence the WOM communication of consumers and, subsequently, influence their purchasing behavior and perceptions of a given brand. Thus, H4-1 is supported by the results of the present study.

Furthermore, studies have indicated that a green brand image positively influences the purchase intention of consumers [92,93]. Gounaris et al. [143] demonstrated that positive WOM contributes to enhancing corporate image. Moreover, consumers who use social media more frequently tend to exhibit more sensitivity toward and identify more with a green brand image, thereby strengthening brand stickiness [43]. According to Leong et al. [144], when consumers spend more time browsing specific products, they tend to prefer these products and make impulsive purchases due to the excitement they experience while browsing the products. Stickiness can attract and retain users, extending the time they spend on an app; thus, it is a crucial metric in social media operations [145]. Therefore, customer stickiness is a major research topic that should be explored in future marketing studies [146]. Our results are in support of H4-2 align with those of other studies, confirming the moderating effect of customer stickiness.

The results of H5 indicate the moderating effect of green concern among the Taipei Fubon Braves fans. However, this moderating effect was not observed among the non-Taipei Fubon Braves fans. The present study confirmed that a green brand image and green WOM have a moderating effect when under the influence of green concern. Social identity theory provides a possible explanation for this finding, positing that the strong sense of identification with a brand among consumers may cause them to follow and participate in activities related to the brand; consequently, they become passionate supporters of the brand, which has become an aspect of their self-concept [147]. Because the non-Taipei Fubon Braves fans did not exhibit this form of brand identification, they might not necessarily follow the brand’s philosophy. Chang et al. [148] suggested that green advertising is more likely to attract the attention of and resonate with consumers if they are highly concerned about environmental issues. In the present study, Taipei Fubon Corporation positioned green concern as part of its brand image. Therefore, through their social identification, the Taipei Fubon Braves fans would likely have endorsed the team’s environmentally conscious behavior and exhibited corresponding behavioral responses, leading to the generation of green WOM (i.e., they recommend the team to environmentally conscious friends). Thus, H5 is supported by the results of the present study.

The present study makes several key contributions. First, by exploring the relevance of establishing a green brand image on TikTok, it highlights the emerging trend where brands use short-form video content on platforms like TikTok. Second, the present study increases our understanding of how professional sports teams can build and shape their green brand image on TikTok, thereby providing empirical support for the development of social media strategies that can help sports teams to positively influence their fans, specifically their acceptance of environmental and sustainability values. Thus, brand managers can develop a more profound understanding of the market and more competently navigate the social media landscape. Through the provision of innovative perspectives, the present study expands the current literature, enriching various concepts in related fields. These key contributions have both practical and academic implications, holding the potential to influence substantial breakthroughs in this field.

## 6. Conclusions

Utilizing the S–O–R theory, this study investigated the interactive relationships among stimuli, organisms, and responses. This framework enabled a nuanced understanding of how social media platforms, particularly TikTok, shape and influence consumer purchase intentions in the context of the green brand image in the digital age. The practical implications of these findings are substantial for businesses aiming to establish a green brand image on TikTok. For brand and marketing professionals, the enhanced understanding gained from this study is crucial in developing targeted strategies to attract and influence potential consumers, thereby gaining a competitive advantage in the market.

This study has several limitations. First, the representativeness of the sample may be limited because the participants were exclusively associated with a specific sport (i.e., basketball). Thus, generalizing the findings to other sports, such as baseball or golf, is difficult. Additionally, for social media platforms, the present study focused exclusively on TikTok, which exhibits distinct features and constraints that are not found in other platforms. Consequently, the results may not be generalized to other platforms or media environments, such as YouTube and Instagram, given the differences in characteristics and audiences among the various platforms. Thus, future studies should consider comparing multiple platforms in order to comprehensively understand the effects of various types of social media on specific topics or phenomena. Furthermore, subjective factors, such as the evaluation of the green brand image, may be influenced by individual subjective differences. Finally, because of the rapid evolution of social media, confirming the long-term effects and changes that participants experience, especially in terms of fan perspectives and loyalty, is challenging. Overall, these limitations should be further considered and addressed in future research.

Future research can expand in several directions to gain a deeper understanding of how professional sports teams can build brand influence on social media platforms. First, a comparative analysis of the effectiveness of green brand marketing strategies across various social media platforms can be conducted, helping to determine the efficacy of these strategies on each of these platforms. Second, exploring the effect of consumer characteristics on the green brand of sports teams can increase the understanding of the needs and values of diverse audience groups. Third, exploring the long-term effect of green brands on consumer loyalty and behavior can provide further insights into how a positive green brand image can affect consumer support and loyalty over an extended period. Fourth, analyzing the influential elements of TikTok content allows for the identification of the most impactful environmental messages and activities. Finally, conducting a comparative study of cross-cultural green marketing strategies can clarify the applicability of environmental messages and strategies in various cultural and geographical contexts. Such research will help businesses to better engage with their audience on social media when it comes to the increasingly central theme of environmental sustainability.

## Figures and Tables

**Figure 1 behavsci-14-00285-f001:**
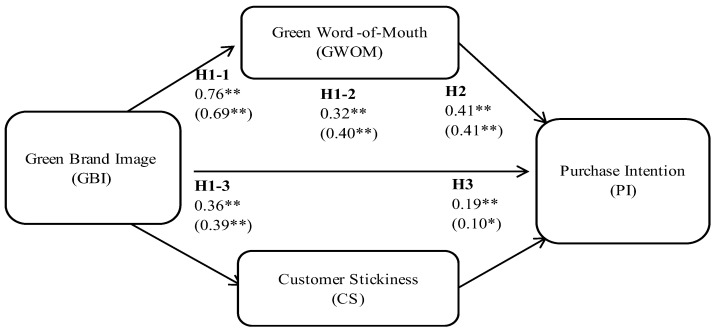
Research framework. * *p* < 0.05, ** *p* < 0.001. The numbers presented outside (inside) the parentheses refer to the non-Taipei Fubon Braves fans.

**Figure 2 behavsci-14-00285-f002:**
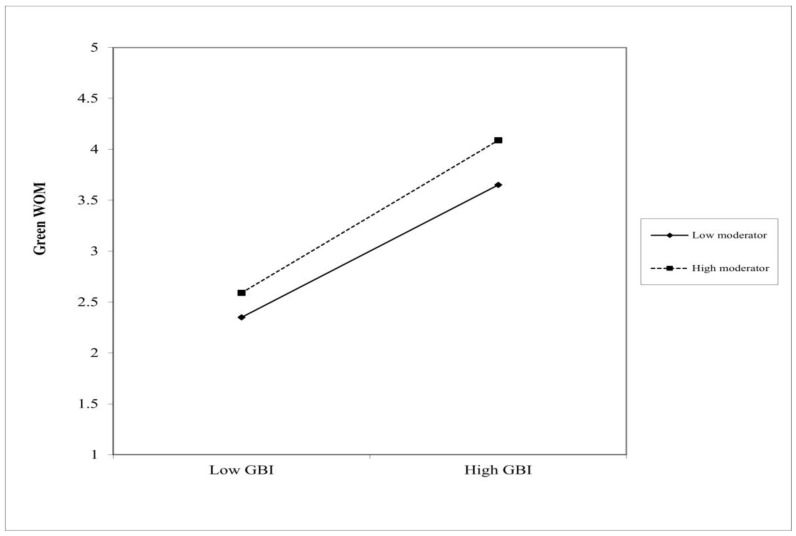
The moderation effects of high and low green concern groups on the relationship between brand image and green word-of-mouth are illustrated in the figure.

**Table 1 behavsci-14-00285-t001:** Summary on Demographic Variables. N_1_ = 298, N_2_ = 302.

Variable		N	%
Gender	Male	138 (154)	46% (51%)
	Female	160 (148)	54% (49%)
Age	20–30 years	73 (71)	24% (24%)
	31–40 years	133 (134)	45% (44%)
	41–50 years	77 (84)	26% (28%)
	51–60 years	11 (10)	4% (3%)
	Over 61 years	4 (1)	1% (1%)
Income (TWD)	<20,000	18 (28)	6% (9%)
	20,001–40,000	128 (129)	43% (43%)
	40,001~60,000	100 (97)	34% (32%)
	60,001~80,000	34 (33)	11% (11%)
	>80,001	18 (15)	6% (5%)
Education level	High school or below	28 (33)	9% (11%)
	Junior college or undergraduate	233 (226)	78% (75%)
	Postgraduate or above	37 (43)	13% (14%)
Marital status	Married	155 (144)	52% (48%)
	Single	143 (158)	48% (52%)

Note. N_1_: sample size for Taipei Fubon fans; N_2_: the numbers presented outside (inside) the parentheses refer to the non-Taipei Fubon Braves fans.

**Table 2 behavsci-14-00285-t002:** CFA results. N_1_ = 298, N_2_ = 302.

Variable	Item	*M*	*SD*	*λ*	*t*
Green Brand Image (GBI) [α= 0.89 (0.91), AVE = 0.63 (0.67), CR = 0.8 (0.91)]					
1. Fubon is considered as the benchmark of environmental commitment		5.36	1.04	0.79	--
		(4.96)	(0.95)	(0.82)	(--)
2. Fubon’s environmental reputation is outstanding.		5.46	1.04	0.84	16.05 *
		(4.93)	(1.03)	(0.84)	(17.19 *)
3. Fubon’s environmental performance is successful.		5.46	1.05	0.80	15.08 *
		(4.93)	(1.05)	(0.85)	(17.53 *)
4. The branding is based on its emphasis on environmental protection.		5.38	1.08	0.75	14.09 *
		(5.05)	(1.08)	(0.78)	(15.29 *)
5. Fubon’s environmental commitment is trustworthy.		5.53	1.04	0.77	14.44 *
		(5.06)	(1.09)	(0.80)	(15.86 *)
Green Word-of-Mouth (GWOM) [α= 0.91 (0.93), AVE = 0.72 (0.79), CR = 0.91 (0.94)]					
1. I would highly recommend this product to others because of Fubon’s environmental image.		5.08	1.22	0.87	--
		(4.58)	(1.30)	(0.90)	(--)
2. I would positively recommend this product to others because of Fubon’s environmental functionality.		5.22	1.16	0.84	18.85 *
		(4.62)	(1.22)	(0.91)	(24.90 *)
3. I would encourage others to purchase Fubon product because it is environmentally friendly.		5.23	1.16	0.86	19.75 *
		(4.69)	(1.27)	(0.90)	(24.21 *)
4. I would say good things about Fubon product to others because of its environmental performance.		5.23	1.12	0.80	17.37 *
		(4.70)	(1.16)	(0.84)	(20.73 *)
Customer Stickiness (CS) [α = 0.94 (0.90), AVE = 0.85 (0.83), CR = 0.94 (0.94)]					
1. I would stay for a long time while browsing TikTok.		4.51	1.53	0.92	--
		(3.97)	(1.70)	(0.92)	(--)
2. I intend to prolong my stays on TikTok.		4.44	1.63	0.92	26.63 *
		(3.72)	(1.68)	(0.93)	(26.44 *)
3. I would visit TikTok frequently.		4.56	1.69	0.92	26.50 *
		(3.84)	(1.82)	(0.90)	(24.74 *)
Purchase Intention (PI) [α = 0.89 (0.90), AVE = 0.67 (0.73), CR = 0.98 (0.91)					
1. I would be happy to purchase Fubon products in the future.		5.33	0.97	0.79	--
		(4.81)	(1.05)	(0.86)	(--)
2. I plan to use Fubon products in the future.		5.36	0.99	0.80	16.01 *
		(4.85)	(1.07)	(0.88)	(19.92 *)
3. I want to take more time to learn about Fubon products because I want to use it in the future		5.40	1.10	0.88	15.76 *
		(4.82)	(1.10)	(0.81)	(17.51 *)
4. I plan to buy Fubon products in the future		5.41	0.99	0.82	16.33 *
		(4.81)	(1.11)	(0.86)	(19.31)
Green Concern (GC) [α = 0.81 (0.86), AVE = 0.55 (0.59), CR = 0.82 (0.85)]					
1. I am worried about the worsening quality of the environment.		5.68	1.15	0.51 *	--
		(5.54)	(1.09)	(0.60 *)	(--)
2. The environment is a major concern for me.		5.55	1.16	0.85 *	8.85 *
		(5.19)	(1.05)	(0.80 *)	(10.46 *)
3. I am passionate about environmental protection issues.		5.55	1.08	0.80 *	8.65 *
		(5.05)	(1.14)	(0.88 *)	(10.98 *)
4. I often think about how the condition of the environment can be improved.		5.43	1.07	0.75 *	8.44 *
		(5.04)	(1.10)	(0.78 *)	(10.27 *)

Note. The numbers presented outside (inside) the parentheses refer to the non-Taipei Fubon Braves fans. * *p* < 0.001.

**Table 3 behavsci-14-00285-t003:** Discriminant validity. N_1_ = 298, N_2_ = 302.

	GBI	GC	GWOM	PI	CS
GBI	0.63 (0.67)				
GC	0.32 (0.28)	0.55 (0.59)			
GWOM	0.56 (0.46)	0.41 (0.36)	0.72 (0.79)		
PI	0.46 (0.50)	0.37 (0.32)	0.53 (0.53)	0.67 (0.73)	
CS	0.10 (0.13)	0.10 (0.06)	0.20 (0.28)	0.22 (0.21)	0.85 (0.83)

Note. Numbers listed along the diagonal denote the AVE values. The numbers listed in the triangle below represent the squared correlation coefficients between the latent factors. The numbers presented outside (inside) the parentheses refer to the non-Taipei Fubon Braves fans.

**Table 4 behavsci-14-00285-t004:** Structural model results. N_1_ = 298, N_2_ = 302.

β	*t*	Results
0.76 (0.69)	12.22 * (12.02 *)	H1-1 supported
0.32 (0.40)	3.80 * (5.83 *)	H1-2 supported
0.36 (0.39)	5.75 * (6.50 *)	H1-3 supported
0.41 (0.41)	5.05 * (6.40 *)	H2 supported
0.19 (0.10)	3.83 * (2.07 *)	H3 supported

Note. Goodness of fit indexes were tested via a structural model; χ^2^/df = 267.690 (267.690)/99 (99) =2.70 (2.70); GFI = 0.90 (0.90); AGFI = 0.86 (0.86); RMR = 0.09 (0.10); SRMR = 0.06(0.06); RMSEA = 0.07 (0.08); NFI = 0.93 (0.93); TLI = 0.94 (0.94); IFI = 0.95 (0.95); CFI = 0.95 (0.95). * *p* < 0.001. The numbers presented outside (inside) the parentheses refer to the non-Taipei Fubon Braves fans.

**Table 5 behavsci-14-00285-t005:** Moderation mediation analysis. N_1_ = 298.

Variable	GWOM (M1)		CS (M2)		PI (Y)	
	B(SE)	95%CI	B(SE)	95%CI	B(SE)	95%CI
GBI (X)	0.65 ** (0.06)	0.76, 0.54	0.41 ** (0.11)	0.63, 0.19	0.31 ** (0.06)	0.42, 0.20
GWOM (M1)					0.32 ** (0.05)	0.41, 0.22
CS (M2)					0.11 ** (0.03)	0.16, 0.05
GC (W)	0.36 ** (0.06)	0.46, 0.23				
X × W	0.10 * (0.04)	0.18, 0.02				
R^2^	0.01					
F	5.46					

Note. * *p* < 0.05, ** *p* < 0.001.

**Table 6 behavsci-14-00285-t006:** Conditional indirect effect at reference group = M ± 1SD. N_1_
*=* 298.

	Effect	SE	95%CI
M-1SD (−0.89)	0.56 **	0.07	0.69, 0.43
M (0.00)	0.65 **	0.56	0.76, 0.54
M + 1SD (0.89)	0.74 **	0.07	0.87, 0.61

Note. The moderating effects of high and low green concern groups on the relationship between brand image and green word-of-mouth.** *p* < 0.001.

## Data Availability

The dataset will be provided upon request.

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
