# Peer review of "Green Consumer Behavior of Sports Enthusiasts on TikTok—An Analysis of the Moderating Effect of Green Concern"

_behavsci, 2024, doi:10.3390/bs14040285_

Round 1

Reviewer 1 Report

Comments and Suggestions for Authors

The title of the article "Green Consumer Behavior of Sports Enthusiasts on TikTok - An Analysis of the Moderating Effect of Green concern", the aim of which was defined as "This study primarily aimed to investigate the effects of green brand image on green word of mouth (WOM), customer stickiness, and consumer purchase intention, with a further examination of the moderating effect of green concern on these relationships".

The theoretical part (literature review) is correct, the hypotheses are also correctly included. In the methodological part, you need to describe more the research procedure and groups (not only in the table, also add what currency is for "Income" in table 1.)

Although it is included in the limitations, it is puzzling whether conducting research on only one sports club (its fans) is sufficient to present the scientific value of this study. It is worth writing down why this club was chosen, whether the survey questionnaire was completed only by people who have access to social media platforms, whether other clubs in his home country also use such activities and what are the global trends.

Reviewer 2 Report

Comments and Suggestions for Authors

The article is meticulously structured and provides a clear exposition of the research purposes, theoretical basis, methodology, findings and conclusions. However, some aspects need to be improved to enrich the understanding and applicability of the results:

Introduction:

-A more detailed exploration of why sports teams, specifically, are a unique context for investigating green branding on social media would enrich the introduction.

-It would be beneficial to detail how TikTok, in its own right, offers unique opportunities for engagement with green brands compared to other social platforms.

Literature review:

-Although the application of the Stimulus-Organism-Response (S-O-R) theory is well founded, it would enrich the work to include a discussion on the specific application of this theory in previous research focused on green brands and social networks, reinforcing its relevance.

-The inclusion of recent investigations that specifically focus on the role of social media in promoting sustainable practices within the sports sector would significantly improve the review.

-The formulation of the hypotheses is logical, but the argument would benefit from more direct connections between the reviewed studies and the proposed hypotheses. For example, when suggesting hypotheses related to green word-of-mouth marketing and purchase intention, direct references to studies that examine these relationships in analogous contexts would be valuable.

-The discussion about TikTok as an emerging platform for these dynamics is introduced, but requires greater depth in explaining the unique characteristics of TikTok that favor the promotion of the green brand image.

Methodology:

-The selection of the Taipei Fubon Braves, due to their distinct commitment to environmental sustainability, adds a new perspective to the research. However, a more detailed explanation of this choice and its implications for the generalizability of the results would expand the scope of the study.

-The sampling and recruitment method through the LINE platform is convenient, but it would be important to recognize and discuss the potential biases of this approach and how they may affect research conclusions.

-A more precise description of the selection and relevance of video content to the hypotheses studied would improve the methodology section.

Results:

-The balanced gender distribution among participants is positive, but a closer examination of how demographic variables such as age, income, education, and marital status may influence the results would provide additional clarity.

-The moderated mediation analysis exploring green concern is profound. Discussing the reasons and implications of the finding that green concern differentially influences the relationship between green brand image and green WOM among Taipei Fubon Braves fans and non-fans would provide valuable insights for marketing strategies.

Discussion:

-The focus on the emotional connection between fans and the green brand, particularly among Taipei Fubon Braves followers, is captivating. Exploring how this emotional connection can be used in marketing strategies would bring practical benefits to brand managers.

-The analysis of the impact of green word of mouth on purchase intentions and its relationship with brand loyalty is enlightening. It would be advantageous to explore strategies to enhance green WOM, especially on social media platforms like TikTok.

Comments on the Quality of English Language

Minor editing of English language required

Reviewer 3 Report

Comments and Suggestions for Authors

The work is adequate, innovative and presents high levels of quality.

Below are the aspects to improve:

The R-squared result for the dependent variable is not presented.

The conclusions section should be structured by presenting the theoretical implications, practical implications and limitations and future research.

Round 2

Reviewer 1 Report

Comments and Suggestions for Authors

Thank you for add to manuscript my comments.

Reviewer 2 Report

Comments and Suggestions for Authors

I am satisfied with the changes made by the authors.